# Progress of Process Monitoring for the Multi-Mode Process: A Review

Jie Ma * and Jinkai Zhang 

Mechanical Electrical Engineering School, Beijing Information Science and Technology University, Beijing 100192, China; 2021020068@bistu.edu.cn
* Correspondence: mjbeijing@163.com

**Abstract:** Multi-mode processing is a central feature of modern industry. The application of monitoring technology to multi-mode processing is crucial to ensure process safety and to enhance product quality. This paper describes the definition, nature and data characteristics of the multi-mode process. A complete classification framework for multi-mode process monitoring methods is produced, covering steady-state modes and transitional processes. After introducing basic concepts and describing research outcomes obtained for monitoring methods, prospects for multi-mode process monitoring technology are discussed.

**Keywords:** multi-mode process; steady-state; transitional process; process monitoring methods

## 1. Introduction

Multi-mode processing is common in modern complex engineering systems and has become a leading production method in industry [1]. In recent years, with the development of automation, it has become possible to collect data continuously from complex non-stationary industrial processes. This data-driven monitoring method can greatly improve the efficiency and applicability of industrial systems through the analysis of data characteristics. Process monitoring refers to monitoring of data that represent the state of the system. Core issues include the prediction of trends in data changes, multi-scale data processing, the elimination of sensor noise, addressing process data uncertainty and the need to establish models that reflect the process of change. The primary goals are to develop systems that can identify important information relating to the state of the production process and to eliminate irrelevant content. Compressing data helps in reducing storage space and in the prediction of changing trends in the data that is processed. According to the operational mode and its scale, modern industrial processes can generally be divided into continuous processes and batch processes [2]. Air conditioning and unmanned helicopters represent examples of continuous processes. An air conditioning system is divided into different mode areas according to the seasons and continuously adjusts to changes in outdoor meteorological parameters [3]. Unmanned helicopters have the capacity for a range of flight modes, including vertical take-off and landing, aerial hovering, coordinated turning, and forward and backward flight and are able to continuously adjust position and posture [4]. Glutamate and penicillin fermentation and injection-molding processes represent examples of batch processes. The glutamate fermentation process is divided into the growth period, formation period and decline period [5]. The fermentation of penicillin includes a rapid bacterial growth stage, a penicillin synthesis stage and a bacterial autolysis stage [6,7]. Injection-molding processes can be divided into three stages: injection, pressure retention and cooling. Traditional monitoring methods used in single-mode processing inevitably lead to inaccurate process performance analysis and false or missed alarms of faults. Effective multi-mode process monitoring, on the other hand, can greatly improve the safety and stability of the equipment production process and reduce production costs.

Therefore, research on multi-mode process monitoring is potentially of great theoretical significance and engineering application value [8].

Ensuring process safety and improving product quality are two urgent problems needing to be solved in modern process industry [9]. In recent years, with the rapid development of science and technology, industrial process systems have become increasingly complex. Multi-mode process monitoring technology, especially widely used data-driven monitoring technology, has received increasing attention from researchers. This paper describes key data-driven multi-mode process monitoring methods and considers future development directions.

## 2. Definition, Data Characteristics and Causes of Multi-Mode Processes

### 2.1. Definition, Data Characteristics of the Multi-Mode Process

1. Multi-mode: multiple operation states and stages in industrial production processes are referred to as multi-mode. Modes and physical phases do not necessarily closely correspond to each other.
2. Multi-mode process: a process with multiple stable mode states in the same production process affected by changes in the external environment and other conditions, or by changes in the production scheme or inherent characteristics of the process itself, is referred to as a multi-mode process. Depending on the different production modes adopted, multi-mode processes can be divided into multi-mode continuous processes and batch processes. A multi-mode system will continuously switch between the following modes: steady-state mode 1—transition mode—steady-state mode 2. The data characteristics corresponding to each mode will be different, such as shifts in the mean value of variables and changes in the covariance structure.
3. Continuous process: a continuous process is a mode that works stably according to the state. Its characteristic variables change slowly.
4. Batch process: batch processes involve a cyclic mode of work. There are multiple steady-state modes in batch processes and their time characteristics change with progress of the process; there are also short-term transition modes in adjacent batch processes.
5. Steady-state mode: a steady-state mode is a mode in which the operating state of the system is relatively stable during production. The variables characterizing the steady-state mode, such as the mean, variance, and correlation, do not change with operation time. In the production process, a system will usually operate in a steady-state mode, which is the crucial mode that determines product quality.
6. Transition mode: when a production process goes from one steady-state mode to another, it cannot do so suddenly. A gradual and dynamic transition mode is required to connect two steady-state modes. In the transition mode, the process variables and operation modes can change greatly over a short period, and the running track is generally different each time. The characteristic variables of the transition mode show strong time-varying, dynamic and non-linear characteristics and have a considerable impact on the production process.

### 2.2. Causes of the Multi-Mode Process

1. Changes in raw material properties, the occurrence of chemical reactions, changes in the external environment, and changes in the operational task and production scheme, can cause the normal mode area of the process to drift over time, such as in chemical distillation processes and other multi-mode processes.
2. Changes in the process parameters or loads, equipment reorganization, and equipment aging, can lead to changes in the relationships between process variables, thus showing obvious multi-mode characteristics. For example, reciprocating mechanical steam turbines show variable working modes during startup, shutdown and load changes. During the operation of mills, steel balls are constantly worn, resulting in

slow changes in modes [10]. With different loads, an engine can respond using several speeds, or different gears to influence power, resulting in multi-mode processes.

3.  The inherent characteristics of a production process mean that it will include multiple operation periods. For example, the injection-molding of plastic products is divided into five operational stages for an injection batch: mold closing, injection, pressure maintenance, cooling and mold opening. It represents a typical batch process [11].

4.  In the start-up and shutdown stages of a process, operation modes, such as mode adjustment and mode switching, inevitably involve a transition mode. The transition mode usually lasts for a short time with less easy access to process data. Therefore, a transition mode will not only require the intervention of many operators but often also results in the production of unassessed products, which can greatly impact both safety of production and product quality [12].

## 3. Multi-Mode Process Monitoring Methods

The identification and classification of multi-mode processes are based on information changes occurring in the production process relating to different modes in the process. Classification is a necessary step for multi-mode process monitoring, evaluation, modeling, quality prediction and process optimization [13]. Whether in continuous or batch processes, a multi-mode process will include both steady-state and transition modes [14]. A complete and comprehensive classification framework for multi-mode process monitoring methods requires to include both steady-state mode and transition mode process monitoring methods. A classification framework for multi-mode process monitoring methods is shown in Figure 1.

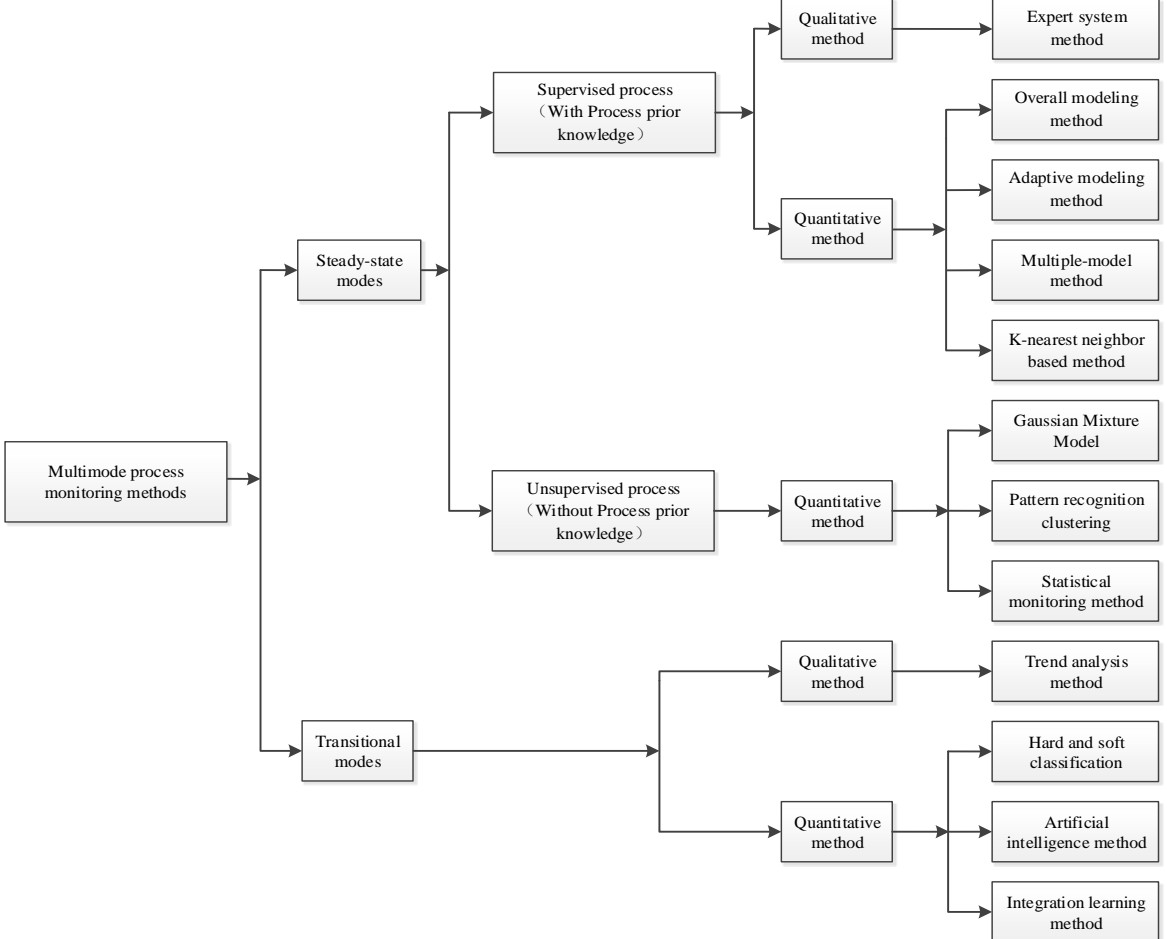

**Figure 1.** A classification framework for multi-mode process monitoring methods.

The main purpose of multi-mode process monitoring methods is to detect and avoid abnormal states during system operation to minimize possible accidents and losses and to ensure the normal and stable operation of systems. The main task of process monitoring is to identify, diagnose and isolate faults. The steps involved in multi-mode process monitoring include fault detection, fault identification, fault diagnosis and process recovery. In the monitoring and identification stage, it is necessary to determine whether the system working state is normal. In the event of abnormalities, the system data needs to be analyzed to identify variables that may lead to failure of the system. The analysis of variable data contributes to locating the fault, assessing the severity of the fault, and then enabling determination of causes. System faults are diagnosed by isolation and troubleshooting. Finally, process recovery is achieved through the action of relevant operators with respect to the fault [15].

## 4. Steady-State Mode Process Monitoring Methods

Steady-state modes are divided into supervised processes and unsupervised processes. If prior knowledge of the process is sufficient (i.e., the mode of the sample data during modeling is known), it is referred to as a supervised process; if not, it is referred to as an unsupervised process.

### 4.1. Supervised Process Monitoring Method

A supervised process is where the mode type and process are known, or can be simply divided by a clustering algorithm. Existing data-based monitoring methods for supervised processes include the expert system method, the overall modeling method, the adaptive modeling method, the multiple-model method and the K-nearest neighbor method.

#### 4.1.1. Expert System Method

The expert system method is a qualitative method that uses human experts to observe the system state, qualitatively evaluates the system state using expert knowledge of the system and faults, and infers the faults that may occur in the system. This method is simple to model but suffers from uncertainty in describing the results and entails consumption of much expert energy. The basic purpose of the expert system method is to establish a knowledge base from engineering experience accumulated by experts over a long period, to design a set of computer programs to simulate the thinking and judgment of experts, and to infer process data information according to the knowledge base. Expert system methods can be classified into traditional expert systems, fuzzy expert systems and belief rule-based expert systems. Lee et al. [16] proposed a multi-mode process monitoring strategy based on an expert system that formulated if-then rules from the knowledge base to enable the identification and monitoring of each mode.

#### 4.1.2. Overall Modeling Method

The overall modeling method considers the multi-mode process as a whole and involves establishing a unified monitoring model for the multi-mode process and building a global monitoring statistical index. Yu et al. [17] proposed a process monitoring method based on global and local models. This method involves learning a global model from historical data, treats multi-mode data as a whole and constructs a global monitoring statistical index to monitor the current process. The authors of [18] proposed an extended principal component analysis (PCA) algorithm by assuming that the covariance matrix of each independent mode had the same eigenvector quantum space, and solved the monitoring problem of the multi-mode process using a modeling method based on a common subspace. Xie et al. [19,20] combined a single-mode support vector data description (SVDD) algorithm with a nearest neighbor difference (NND) algorithm and an improved local neighbor standardization (ILNS) algorithm. After preprocessing the original data, eliminating the multi-model characteristics and converting them into single-mode data using NND, they constructed an SVDD monitoring model. The control limit of data statistics for

monitoring and solving the problem was calculated, as the traditional SVDD algorithm was unable to deal with multi-mode data.

The overall modeling method is simple and easy to implement because it does not need to model each mode separately. However, it can only be used when the data of each mode do not differ greatly and the non-linear relationship between variables is not strong as, otherwise, the overall modeling will be difficult to describe uniformly. Due to these limitations, the method is rarely used in practice.

### 4.1.3. Adaptive Modeling Method

The adaptive modeling method continuously adds new process data to the modeling data matrix to make algorithms track new process modes and update the monitoring model in real-time. Li et al. [21] proposed a recursive and iterative PCA model that adapts to changes in modes by adding new sample data to the model through a recursive and iterative algorithm. Based on external analysis, the authors of [22–24] introduced adaptive monitoring, which decomposes the process variables into main and external variables, and gradually eliminates the influence of external variables using a least square regression algorithm to perform process monitoring. The authors of [25] proposed an adaptive monitoring method based on a hierarchical clustering structure, assuming that each mode has the same covariance structure. However, this assumption greatly limits its application. Choi et al. [26] also proposed an adaptive monitoring method to obtain a new regression model by updating the mean and covariance structure in real-time. Cheng and Ge et al. [27,28] combined just-in-time learning (JITL) with PCA and least square support vector regression (LSSVR) and used JITL to model the residual sequences formed by the differences between the predicted value outputted from the process and the sample data, thus realizing real-time monitoring. Huang et al. [29–31] introduced the idea of applying an adaptive dictionary learning algorithm to multi-mode process monitoring. The original data is classified and trained by dictionary learning, and the K-singular value decomposition (KSVD) algorithm is introduced to process the dictionary. After the training data is sparsely represented by the dictionary, the model is constructed for monitoring and eliminating the mode noise. The limitation of this method is that the modal data can be used by default. Xiao et al. [32] proposed an adaptive process monitoring method, called distributed adaptation local outlier factor (DALOF), to improve the performance of monitoring statistics. The method divides the source sample training data into the source and target domains, weights them, and verifies the monitoring accuracy by comparing the maximum mean discrepancy (MMD) between the source domain and target domain. This method is suitable for complex processes with non-linear characteristics.

Generally, the adaptive modeling method is suitable for process monitoring in the case of slow mode changes (i.e., "slow time-varying"). This method involves heavy modeling data and computation demands, causing a response lag of the monitoring model to the current monitoring sample and cannot reflect the state of the current sample in time; it also depends on a mechanism model and prior knowledge of the process. Therefore, it is not frequently applied in practice.

### 4.1.4. Multiple-Model Method

Since the mean, variance and correlation of process variables under various modes are different, it is difficult to describe them with a unified model. The multiple-model method establishes sub-models for each mode, then designs a pattern-matching rule and selects the best matching model for monitoring. Zhao et al. [33] first divided the sample data into multiple modes, based on prior knowledge of the process, then established PCA models for each mode, and finally selected the model with the minimum squared prediction error (SPE) statistics for monitoring. Guo et al. [34] established PCA models for training data in multi-modes, determined the matching coefficient of the monitoring control limits in each mode by selecting common multiple methods, and formulated a unified control limit for monitoring.

Another aspect of the multiple-model method is the establishment of sub-models for each mode and the construction of a global statistical index to monitor the multi-mode process. The authors of [35–37] proposed a multiple-model method based on Bayesian theory and constructed a global probability index for monitoring. Xie et al. [38] proposed an integrated monitoring index $D_{global}$ integrating the Mahalanobis distance in multi-modes, which can describe the internal statistical information of each mode and include global information for all modes.

In the actual industrial environment, most of the parameters in the multiple-model method change with the system state during monitoring and monitoring ability will not be lost as a result of failure of one of the methods. The method has great advantages over the overall modeling method and has been widely used in recent years with fruitful research results.

### 4.1.5. K-Nearest Neighbor (KNN) Based Method

The K-nearest neighbor method is often used as a multi-mode process monitoring statistic to judge the system state by observing the distance relationship between sample data and their K-nearest neighbors. He et al. [39] proposed fault detection method using the K-nearest neighbor (FD-KNN) method based on K-nearest neighbor multi-mode process monitoring. Taking the square sum of the Euclidean distance between sample data and their K-nearest neighbor $D^2$ as the monitoring statistics index, this method can increase the computational burden of the system. Additionally, when the variance structure of multi-mode data is different, the statistical index cannot correctly reflect the differences between samples, resulting in false or missing reports. Guo et al. [40] constructed K-nearest neighbor monitoring statistics in the residual space to improve the fault resolution of the system. Zhang et al. [41] replaced the original Hotelling's $T^2$ monitoring statistics with the monitoring statistics of K-nearest neighbors and the Mahalanobis distance of kernel entropy projection principal component space to improve the rate of early warning and accuracy of fault monitoring. Feng et al. [42] defined the monitoring statistics as the mean of the sum of squares of the original distance $D^2$ based on the FD-KNN algorithm to improve the monitoring performance in case of large modal differences.

To optimize the calculation amount of the traditional KNN and improve the monitoring accuracy, Zhang et al. [43] proposed a variable moving window K-nearest neighbor (VMW-KNN) method. A variable-length sliding window is used to determine the K-nearest neighbor search range of the sample, and then KNN is used to preprocess the sample data to calculate the load matrix. Finally, the corresponding sub-PCA model is selected for monitoring. Zhong et al. [44] combined kernel density estimation with the K-nearest neighbor algorithm to construct a local density function, i.e., density estimation of K samples, reducing the sensitivity of window width parameters and eliminating multi-modal differences in data. Zhang [45] proposed a principal component difference method based on the K-nearest neighbor to obtain the K-nearest neighbor augmented matrix of each sample. The loading and score matrices using PCA were then used to calculate the data sample, estimate scores and monitor the system by the new monitoring statistics determined by the covariance matrix of the differences between them.

To improve the variance structure difference of data, Feng et al. [46] proposed a principal component and weighted K-nearest neighbor (PC-WKNN) method and obtained the weighted distance statistic $D$ to reduce the variance difference of data by calculating the distance between the training sample and the K-nearest neighbor and the reciprocal of its local nearest neighbor in the principal component space. Guo et al. [47] proposed a KNN based on local relative probability density (LRPD-KNN), which combined KNN with kernel density estimation, to establish a local probability density estimation function, and used the local relative probability density to improve data variance.

In addition, Ma et al. [48] proposed a multi-mode process monitoring method based on distance space statistics analysis (DSSA). This method detects the presence of a fault at the current time by analyzing the difference between the K-nearest neighbor distance of

normal samples and fault samples. The method is suitable for cases where the variance structure of data is quite different in different modes.

### 4.2. Unsupervised Process Monitoring Method

An unsupervised process occurs where the prior details of the process are unknown or are insufficient. Existing data-based monitoring methods for unsupervised processes include the Gaussian mixture model (GMM), the pattern recognition clustering method and the monitoring statistics method.

### 4.2.1. GMM

GMM is a weighted combination of multiple single Gaussian probability density functions, the number of which is equal to that of the modes. The GMM shows excellent noise processing ability and robustness in dealing with motion trajectory problems, and, as a result, is widely used in motion graphics processing [49]. Additionally, this model is used to divide the Gaussian component of data to determine the confidence limit and has been applied to fan icing monitoring [50]. The GMM does not rely on prior knowledge and can automatically obtain multiple Gaussian model information through the learning and training of process data. However, its ability to extract non-Gaussian data is weak.

There are several overall modeling and multiple-model modeling approaches to GMM modeling. When all modes obey Gaussian distribution, an overall GMM can be established for the process. Choi et al. [51,52] proposed a process monitoring method combining GMM with PCA and Fisher discriminant analysis. A GMM was first established in the low-dimensional feature space, and then PCA was used for process monitoring. Zou et al. [53] proposed a non-optimal state cause tracing method based on the partial derivative of variables combining GMM and Bayesian reasoning. A unified GMM was established for the data of the same operating state, and a monitoring control limit was constructed through the partial derivative contribution index of dominant variables with great state influence to judge the non-optimal cause. This method avoided modal misclassification. Guo et al. [54] established an improved variational inference Gaussian mixture model based on locality preserving projection (IVIGMM-LPP). The model preprocesses data using locality preserving projection (LPP), introduces the eigenvalue discrimination condition of the covariance matrix and the standard deviation vector to reduce the influence of initial parameters and unify data, and builds an overall GMM model for monitoring. However, this model is not suitable for monitoring small faults. Gao et al. [55] used an expectation-maximization (EM) algorithm and Bayesian Yin-Yang (BYY) algorithm to estimate the parameters and number of modes in the GMM, respectively. They also used PCA to reduce the dimensions of the data and built a multi-mode GMM to improve the monitoring performance of the multi-mode process. Tang et al. [56] proposed a Gaussian mixture variational autoencoder (GMVAE) model. Using a latent variable space projection method, the model, which is combined with a multi-variable normal distribution model and a variational posterior model, makes the modes obey the Gaussian distribution. Overall modeling requires that the covariance structure of each mode data should be consistent, otherwise, the covariance of the process data will be strange. Evidently, overall modeling has significant limitations.

Tan et al. [57] established the GMM for different stable and transition modes for process monitoring. Xie et al. [58] combined LPP and a multi-model GMM to monitor the multi-mode process. The LPP algorithm was used to reduce the dimensions of multi-mode data, then the GMM was established for each mode in low-dimensional subspace, and the local probability index was integrated into the global Bayesian rule probability index for monitoring.

### 4.2.2. Pattern Recognition Clustering Method

The performance of process monitoring largely depends on the clustering effect of sample data. The pattern recognition clustering method first introduces various clustering

analysis methods in pattern recognition into process monitoring and then combines them with other monitoring methods for process monitoring. The clustering method tends to be limited to a predetermined number of clusters, which is difficult to realize in practice.

K-Means Clustering Method

The K-means clustering method involves first randomly classifying the data, taking the mean of all samples in each cluster subset as the center of the category, then classifying the data according to each classification center, and then iterating continuously until the cluster center converges to determine the category of each sample. Niu et al. [59] used the K-means clustering method and multi-PCA model to monitor the process of variable modes. First, the modes were divided by the K-means clustering method, and then a multi-PCA model was established for each mode for process monitoring. This method has been successfully applied to monitoring the boiler process in power plants. Zhang et al. [60] used the K-means clustering method to monitor multi-stage batch processes, and it has been successfully applied to the process monitoring of penicillin fermentation and semiconductor manufacturing. Zhang [43] used the sliding window cutting data matrix to calculate the mean value, then applied the improved K-means to cluster the window mean vector to realize the fully automatic monitoring of the multi-mode. The results of the K-means clustering algorithm can be greatly affected by the initial classification and its monitoring effect is unstable.

To reduce the impact of other data on the clustering process, the K-center point clustering method is used to construct data clusters by iterating the data. After randomly selecting the center point of each cluster and calculating its distance from other data, the data are assigned to the nearest center point and iterated continuously until each data point is the same as the corresponding center point. Liang et al. [61] improved the distance calculation approach of the K-center point algorithm using the shared nearest neighbor distance combined with the dynamic kernel principal component analysis method for the online monitoring of fixed-wing UAVs in multi-mode.

Fuzzy C-Means Clustering (FCMC) Method

Proposed by Bezdek in 1981, FCMC is used to calculate the C clustering centers and the membership function matrix of the sample set by a recursive iteration method according to the clustering criterion function and to determine the category of the sample set by membership [62]. Wang and Liu et al. [63,64] proposed an FCMC method and a distance-based fuzzy C-means clustering model (DFCM) method, respectively. The methods have been successfully applied to the process monitoring of ethylene production and cracking furnaces. Luo [65] calculated the number of clusters through singular value decomposition and a K-means clustering algorithm, used FCMC to divide the process modes into steady-state and transition modes according to the membership degree, and established monitoring statistics according to modes. Xie et al. [66] combined LPP and DFCM to monitor the multi-mode process by first reducing the dimensions of the historical data and then dividing the modes by clustering. This enabled retention of all clustering information and improved sensitivity to outliers.

Mixture Modeling (MM) Method

The MM method is used to estimate parameters by modeling each cluster using an EM algorithm and determining the number of modes in the multi-mode process. Xu et al. [67,68] used an independent component analysis mixture model (ICAMM) based on Bayesian estimation to divide and cluster the data and introduced Bayesian reasoning to build monitoring statistics to monitor the data. This method is demanding in the selection of data parameters. Jiang et al. [69] proposed a modeling method referred to as a variable Bayesian Gaussian mixture model with canonical correlation analysis (VBGMM-CCA). The method uses VBGMM to identify the number of patterns of sample data, establish residuals, cluster the training data in each pattern and construct a local canonical correlation analysis (CCA) model. Finally, the parameters of each local model can be unified to establish a unified Bayesian inference probability (BIP) index monitoring statistic. This method can automatically identify the number of operation modes in certain modes.

### 4.2.3. Statistical Monitoring Analysis Methods

The monitoring statistics analysis method assumes that the sample data is normal data that meets all requirements and starts by analyzing the multivariate spatial distribution characteristics between normal data and fault data. It then establishes the corresponding monitoring statistical indicators by analyzing and comparing the distance relationship between them. The traditional methods often take Hotelling's $T^2$ value, which measures data changing in the main space, and the squared prediction error (SPE) value projected on the residual subspace, as the monitoring statistical indicators—if there is significant change, a fault is considered to have occurred. This method does not need to cluster the historical modeling data and can still effectively monitor the multi-mode process in the absence of prior knowledge of the process.

Multivariate Statistical Process Control (SPC) Methods

Based on the multivariate statistical analysis method of statistical control charts, a traditional Hotelling $T^2$ monitoring statistic distribution of values is proposed on the assumption that the covariance matrix relates to unchanged conditions. Sample data, from high-dimensional space projection to the low-dimensional feature space, is used via statistical measurement and control to improve product quality and maintain the primitiveness of data as much as possible. Multivariate statistical process control charts are mainly divided into $T^2$ control charts, that control the mean vector, and Wishart distribution $\Lambda$-type $F$ control charts of control covariance matrix promoted by univariate statistical $x^2$ distribution control charts [70]. The SPC method is widely applied and can be combined with a time series model or an artificial intelligence method to produce a classification control chart method to improve algorithm performance under the assumption of data nonlinearity [71,72].

Local Density Factor Based Method

Guo et al. [73] proposed a local neighborhood standardization-local density factor (LNS-LNF) method. The local neighborhood standardization (LNS) method is used to preprocess the data to conform to a Gaussian distribution, calculate the local density factor value, and determine the control limit of monitoring statistics by combining the kernel density estimation method to judge the sample state. Xu et al. [74] proposed a global probability index based on local density to replace the traditional monitoring index. The local density in the dataset makes use of the local nearest neighbor features rather than relying on the data distribution, which can better reflect the actual engineering modes. However, the method based on LNS may misjudge the data under new operation models.

Based on a local density factor method, some researchers have proposed a local outlier factor method which defines the fault sample as an outlier factor and assesses whether the training sample is in a fault state by comparing the differences between the training sample and the normal sample. The authors of [75,76] screened the nearest neighbor samples meeting the distance requirements using weighted distance or Euclidean distance, calculated the local outlier factor by calculating the reachable distance and local reachable density, and combined these to calculate the monitoring control limit to improve monitoring accuracy.

## 5. Transition Modes Process Monitoring Method

Existing process monitoring methods of transition modes are mainly divided into qualitative and quantitative trend analysis methods, such as the artificial intelligence method, hard classification and soft classification, and the integrated learning method.

### 5.1. Trend Analysis Method

In both continuous production and batch processes, process variables show a dynamic and gradual trend in transition mode. The library trend is qualitative, indicating the change trend in process variables under normal operation, which can be obtained from the historical data of the transition mode. The real-time trend of process variables is obtained from online process data. When a fault occurs, the process variables show a trend in the

fault characteristics. Anomalies are identified by qualitatively comparing real-time trends with library trends. Based on a traditional qualitative trend analysis method, Ref. [77] selected the shape of the trend as the qualitative process variable and added quantitative information, such as the amplitude and duration of the trend, and put forward a dynamic feature synchronization algorithm. This method has been applied to the testing of semi-industrial distillation towers and simulated fluid catalytic cracking units to accurately detect faults.

### 5.2. Hard Classification and Soft Classification

Hard classification and soft classification are defined according to whether the sub-stage is strictly separated from the process by 0 or 1 in the transition data clustering membership. Lu et al. [78] proposed a "hard classification" method to determine the transition process among steady-state modes. In this method, two adjacent sub-period models of the process are weighted to form an overall model and approximately describe the characteristics of the transition mode.

To better classify the transition and stable regions, Zhao et al. [79,80] further proposed a "soft classification" method with the batch process as the research background. In this method, the degree of 0–1 fuzzy membership is used as the weight coefficient of the two sub-period models adjacent to the transition modes and the sub-period is weighted and fused.

Both hard and soft classification methods are poor in processing ability for multi-mode batch processes due to a lack of data information on transition modes.

### 5.3. Artificial Intelligence Method
#### 5.3.1. Neural Network Method

Ng et al. [81] proposed a neuron-clustering method; the online data of the transition mode were mapped into a track by self-organization mapping (SOM) and transformed into a series of neuron clusters, which represented the characteristics of the operating state. The monitoring of the transition mode was realized by comparing it with the known characteristic sequence of the transition mode in the knowledge base. This method realized the visualization of the operating state in the transition mode with rapid online application speed. Han et al. [82] projected the source and target domains into the sub-feature space through an adaptive layer. They also introduced a convolutional neural network (CNN) into a joint domain adaptive model to reduce the maximum mean difference (MMD) and to improve the feature distribution difference of the adaptive layer as well as the migration ability. Additionally, CNN can be divided into the sharing layer and the full connection layer (adaptive layer). Ma et al. [83] proposed a method based on a deep belief network (DBN) optimized by multi-order fractional Fourier transform (FRFT) and a sparrow search algorithm (SSA). Firstly, the FRFT, based on curve feature segmentation, was used to extract the fault characteristic frequency of the system vibration signal. Then the SSA was used to iteratively optimize each weight parameter of DBN the maximum number of times, which was used as the initial weight of the model, train the SSA-DBN model together with the fault characteristics, and apply it to fault monitoring.

#### 5.3.2. Autoencoder Model-Based Method

As a classical method of the deep neural network, the depth learning monitoring method based on autoencoder projects the data samples into the output space through encoding and decoding steps. LV et al. [84,85] proposed a representation learning adaptive monitoring method based on a stacked sparse autoencoder (SSAE). The SSAE model is constructed by training the uppermost autoencoder to the latent layer as the input of the next encoder for multiple superpositions and is combined with the autoencoder network to monitor the transition mode. Jiang [86] proposed a dynamic sparse stack autoencoder model (DSSAE) which used an unsupervised data dynamic expansion matrix to extract the dynamic characteristics of DSSAE and to improve the monitoring and classification ability

of the system by training the model with a back-propagation (BP) algorithm and gradient descent method.

To monitor the non-linear mode, Lee et al. [87] constructed a variational autoencoder (VAE) model, which is composed of encoders and decoders with multilayer neural networks. The encoder reduces the dimension by mapping the dataset to the latent space, and the ReLU function is used as the activation function. The decoder maps the latent vector to the output space. Wu et al. [88] proposed an adaptive monitoring method based on local adaptive standardization and variational autoencoder bidirectional long short-term memory (LAS-VB). They preprocessed local moving window data for standardization by the LAS method, constructed a VAE model composed of multiple bidirectional long-term and short-term memory layers, and trained the model to calculate the standardized data to obtain the monitoring threshold and to assess the system state. The model has a strong ability to deal with the new mode, but it cannot handle slightly variable drift faults or distinguish the difference between normal deviations and faults. Ma et al. [89] used a variational automatic encoder and random forest information fusion method for bearing fault diagnosis and residual life prediction.

### 5.3.3. Dirichlet Process-Based Method

To effectively adapt to the new mode, some researchers have selected Gaussian mixture data following the Dirichlet process as a sample to calculate the distribution. A Dirichlet process-Gaussian mixture model (DP-GMM) was constructed after obtaining the distribution mean and covariance matrix and iterating the sample parameters for cluster analysis of non-linear data [90–92]. This method is essentially an unsupervised clustering algorithm. A disadvantage lies in that its distribution initialization stage may cause parameter errors, and it is sensitive to the monitoring of modes in the transition period, which may lead to misjudgment.

### 5.3.4. Hidden Markov Model (HMM)-Based Method

The HMM is composed of Markov chains and state transition matrixes and contains four key components: hidden state, state transition probability, observable state and initial state. The model treats each mode as a hidden state by default, but the duration of the state is required to obey a geometric distribution. This problem can be overcome by the hidden semi-Markov model (HSMM), constructed by introducing the probability of state duration based on HMM. If the HMM transition probability prior distribution is defined using the multi-layer Dirichlet process hybrid model to cluster the data, the HDP-HSMM method constructed with Mahalanobis distance as the local monitoring statistic can automatically identify and classify the data mode after using the HSMM to monitor the mode conversion in the transition mode [93]. After being trained by the Baum–Welch method, HMM can be combined with kernel principal component analysis (KPCA), or variable selection methods and moving windows, to significantly improve the monitoring performance of transition modes and reduce rates of false alarm [94–96].

### 5.3.5. Just-In-Time Learning (JITL) Method

The JITL method comprises a stacked autoencoder (SAE) and CCA neural network. In deep canonical correlation analysis (DCCA), it is divided into a joint learning process of dividing historical data, extracting input and output, constructing SAE personal learning and optimizing SAE parameters to improve parameter relevance. A CNN is introduced into the two processes to construct a residual monitoring mode. The monitoring statistical limit can be obtained by selecting and estimating historical data from online data. The model can be used to monitor the complex non-linear mode, but it may cause misjudgment due to the impact of small interferences on data monitoring statistics [97,98]. Chen et al. [99] proposed an improved JITL-CCA method. They constructed a CCA model through K-means to classify the minimum input-output-related dataset obtained from the training data. After removing the omitted data, the CCA model was reconstructed. The parameter matrix was

calculated, and the statistical limit was monitored. This algorithm is poor in dealing with new operation modes and faults due to its high dependence on learning datasets.

To realize the division of phase and transition, Zhang et al. [100] combined statistics pattern analysis (SPA) and slow feature analysis (SFA) models to form statistics slow feature analysis (SSFA). The most relevant samples were selected from local data by a JITL algorithm to construct a similarity matrix and monitor the current test samples. Zhang et al. constructed a global preserving SSFA model based on real-time learning by introducing global analysis and local statistical matrixes to shorten the monitoring time of abnormal states and improve monitoring accuracy.

### 5.4. Integrated Learning Method

To extract the characteristics of Gaussian and non-Gaussian information in the data for more accurate mode classification in multi-mode process monitoring, a dynamic integrated clustering method based on the kernel ICA-PCA (K-ICA-PCA) model [101,102] was used for the modeling of the transition mode. Then a transition mode process monitoring method was adopted based on PCA feature extraction and multi-class SVDD pattern classification. The effectiveness of the algorithm was verified by the Tennessee Eastman (TE) process. However, when the algorithm has strict requirements on the selected initial value, an inconsistent initial value may cause a serious loss in its monitoring performance. Li et al. [103] constructed a serial integrated PCA-kernel PCA-serial PCA (PCA-KPCA-SPCA) model based on the common variable subspace and specific variable subspace decomposed by data to improve the non-linear multi-mode monitoring. The limitation of this method is that it ignores other mixing characteristics of the data. Guo et al. [104] proposed a serial integration dynamic-inner PCA-PCA-KPCA model. The loading matrix and latent variable matrix were obtained by the dynamic-inner PCA iterative data matrix to construct a vector auto-regressive (VAR) model and extract dynamic features. The residual singular values of the model were decomposed to obtain the linear features using PCA. Then, the residual features were decomposed to extract the non-linear features using KPCA. Finally, the extracted features were integrated to construct a unified monitoring statistic.

### 6. Conclusions

At present, although valuable research results have been achieved in multi-mode process monitoring technology, there remain many difficult problems worthy of further research: (1) Data acquisition is difficult. In particular, the amount of historical data available in the transition mode is far less than that in the steady-state mode. The process monitoring of transition modes is still a challenge; (2) The mechanism of the batch process is complex, and the model is difficult to obtain. The monitoring technology for the batch process is far behind that for the continuous process; (3) The monitoring of complex industrial production processes in the multi-mode is divided into offline modeling and online monitoring stages. The dilemma for pattern recognition of offline data lies in the means of distinguishing modeling data for steady-state modes and transition modes, while that of online data is how to identify the mode category corresponding to online data in time.

In conclusion, there have been many research studies concerning multi-mode process monitoring technology. However, few studies relate to fault prediction and residual effective life prediction techniques which have significant development potential.

**Funding:** This research was funded by the National Key Research and Development Program of China (No. 2019YFB1705403) and the National Natural Science Foundation of China (No. 61973041).

**Institutional Review Board Statement:** Not applicable.

**Informed Consent Statement:** Not applicable.

**Data Availability Statement:** Not applicable.

**Conflicts of Interest:** The authors declare no conflict of interest. The funders had no role in the design of the study; in the collection, analyses, or interpretation of data; in the writing of the manuscript, or in the decision to publish the results.

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
