# Peer review of "Progress of Process Monitoring for the Multi-Mode Process: A Review"

_applsci, doi:10.3390/app12147207_

Round 1

Reviewer 2 Report

This paper reviews process monitoring methods of multimode processes. The following issues should be carefully addressed before publication:

-It is not clearly explained what a multi-mode process is, especially its difference from a multi-stage process (line 34).

-Please explain why the role of statistical control charts in this review paper is ignored. There are many researches that have used them in residual monitoring and to control cascade effects in a multi-stage process.

-The authors are advised to consider replacing the two paragraphs in the Introduction section.

-It is not explained throughout the text what 'steady-state' and 'transition' mode processes monitoring methods are. Also, in Figure 1 the term 'stable modes' is used instead of 'steady-state' mode.

-Are not sections 4.2.2.1 and 4.2.2.2 talking about the same thing? It looks like they are both describing the applications of K-means clustering methods with only some minor differences.

-What is SPE in section 4.2.3? Also, this section seems to fit the applications of statistical control charts (which are not by the way limited to only T^2 Hotelling control charts)!

-In section 4.2.3.1, why KNN methods are classified as unsupervised methods, considering they are supervised methods?

-In some cases, the topics are not well explained or not explained at all and the authors jumped straight to the researches done related to that topic instead of explaining it well enough first (for example in section 5.3, you should first explain what you mean by hard and soft classifications and then introduce the related works. Please fix this for other sections too, as there are some more similar situations.

Round 2

Reviewer 2 Report

The paper has been significantly revised. However, it looks like they have rushed it, and still, some serious issues remained.

-The English writing, especially regarding the newly added material needs serious polishing.

-Headlines numbering is wrong after Section 2.

-The definitions are still unclear in some cases. For instance, when you want to define 'mode', you say: 'it is the different operation modes of...'. You cannot use the same word for its definition! How can you write a literature review about multi modes when you cannot clearly describe what the bloody mode is?! Also, change all the definitions in a way that they are easily understandable by the readers. This is not the case in many of your definitions.

-The first time you use any abbreviation, describe what they stand for! For example, SPE is not described.

-You say you have replaced the two paragraphs in Section 1, but how come I cannot see it?

-You say statistical monitoring methods belong to the unsupervised section, in which there is no prior knowledge of the process, but without prior knowledge of the process how the statistics and their distributions can be estimated?

-You have only mentioned one reference for section 4.2.3.1 and from what I can see it is in Chinese! Was not there any other paper? 
